# Cadmium Exposure in Aquatic Products and Health Risk Classification Assessment in Residents of Zhejiang, China

**DOI:** 10.3390/foods12163094

**Published:** 2023-08-17

**Authors:** Yue He, Hangyan Fang, Xiaodong Pan, Bing Zhu, Jiang Chen, Jikai Wang, Ronghua Zhang, Lili Chen, Xiaojuan Qi, Hexiang Zhang

**Affiliations:** 1Department of Nutrition and Food Safety, Zhejiang Provincial Center for Disease Control and Prevention, Hangzhou 310051, China; yhe@cdc.zj.cn (Y.H.); 96zhubing@163.com (B.Z.); jchen@cdc.zj.cn (J.C.); jkwang@cdc.zj.cn (J.W.); rhzhang@cdc.zj.cn (R.Z.); llchen@cdc.zj.cn (L.C.); hxzhang@cdc.zj.cn (H.Z.); 2Hangzhou Linping District Center for Disease Control and Prevention, Hangzhou 311100, China; fhy964590@163.com; 3Department of Physical-Chemistry, Zhejiang Provincial Center for Disease Control and Prevention, Hangzhou 310051, China; xdpan@cdc.zj.cn

**Keywords:** Cadmium, aquatic products, health risk assessment

## Abstract

Cadmium (Cd) pollution of food safety is a prominent food safety concern worldwide. The concentration of Cd in six aquatic food categories collected from 2018 to 2022 was analyzed using inductively coupled plasma mass spectrometry, and the Cd exposure levels were calculated by combining the Cd concentration and food consumption data of 18913 urban and rural residents in Zhejiang Province in 2015–2016. The mean Cd concentration was 0.699 mg/kg and the mean Cd exposure of aquatic foods was 0.00951 mg/kg BW/month for the general population. Marine crustaceans were the largest Cd contributor, corresponding to 82.7%. The regional distribution results showed that the average Cd exposure levels of 11 cities did not exceed the provisional tolerable monthly intake (PTMI). According to the subgroups, the Cd mean exposure level of 2–3-year-old children was significantly higher than that of the other age groups but did not exceed the PTMI. Health risk classification assessment demonstrated that the final risk score was six, and the health risk level of Cd exposure in aquatic products in the Zhejiang population was medium. These results demonstrated that the risk of Cd exposure in certain food types or age groups should be given more concern.

## 1. Introduction

Cadmium (Cd) is a nonessential transition and toxic metal that poses severe threats to human health and may be associated with numerous diseases. Cd can enter the body through water, soil, air, diet, tobacco, and occupational exposure and has a long half-life of approximately 10–30 years [1]. Important anthropogenic Cd sources include mining, atmospheric deposition of combustion emissions, and the use of Cd-containing fertilizers [2]. Cd is an extremely toxic heavy metal, and there is no specific, safe, and efficacious therapeutic management for Cd toxicity [3]. Epidemiological evidence has revealed that environmental and occupational Cd can be associated with various types of cancer, cardiovascular disease, diabetes, chronic renal disease, osteoporosis, and Alzheimer’s disease [4,5,6]. Recent research has elucidated that Cd acts as an immunotoxic agent, modulates the immune system, and responds [7,8]. Cd toxicity as an important health hazard which may compromise host defense against pathogenic microorganisms and homeostatic reparative activities. Cd exposure triggers oxidative stress, necroptosis, and Th1/Th2 imbalance, and promotes inflammation through the TNF-α/NF-κB pathway [9]. Moreover, Cd affects cell proliferation, differentiation, apoptosis, and other cell activities at the cellular level [10].

Due to a relatively high volatility, large ionic radius, and chemical speciation, Cd is susceptible to mobilization by anthropogenic and natural processes [11]. Cd in the environment can be enriched by foods and then transferred to human beings by the food chain. In 2011, the joint FAO/WHO Expert Committee on Food Additives (JECFA) formulated the health guideline value (HBGV) of Cd, which states that the daily intake of Cd for adults should not exceed 50 μg, and the provisional tolerable monthly intake (PTMI) for Cd was 0.025 mg/kg body weight (BW) [12]. The European Food Safety Commission (EFSA) set an HBGV of no more than 21.6 μg/day [13]. The National Health and Nutrition Examination Survey (NHANES) 2007–2012 showed that dietary exposure to Cd in the American people was 4.63 μg/day [14]. A Northern Italian dietary study reported that the estimated median dietary Cd intake was 5.00 μg/day, which far exceeds the daily dietary intakes recommended by national international agencies [15]. In China, especially in the southern area, some foods contain Cd that exceeds the Chinese food limits, which are limited from 0.003 mg/kg in natural mineral water to 3 mg/kg in sea crab [16,17]. Compared to other standards worldwide, the food limits of Cd are 0.005–3 mg/kg in the European Union, 0.05–2.5 mg/kg in Australia, and 0.003–2 mg/kg in the international Codex Alimentarius Commission (CAC), respectively. A health risk assessment in China’s Guangdong Province indicated that rice grain, leaf vegetables, bivalves, and shrimp and crabs were the main contributors of Cd exposure [18].

Generally, the quality of water is compromised due to the contamination of toxic metals, which adversely impacts the health of humans, animals, and plants. A risk assessment of 87 Chinese lakes from 2009 to 2019 indicated that 5.6% of lake water and 88.2% of lake sediments were polluted by Cd [19]. The survey of 108 sediment samples from the Bohai Sea showed that the Cd concentrations were considerably higher than the average upper crust value and presented high potential ecological risk [20]. Aquatic products from fresh water or the ocean present numerous health benefits but also accumulate toxic metals and metalloids, which can have harmful effects on human health if consumed in excess [21]. Fish, crustaceans, and bivalves are the most common aquatic foods and are also significant sources of Cd exposure [22]. These aquatic products can accumulate Cd in their tissues, undergo osmotic exchange and ingestion at the body surface, and absorb Cd from water [23]. A study from Laizhou Bay in China revealed that Cd was the second most abundant metal and the most accumulated element in seafood, and crabs showed a much higher enrichment ability of Cd than others [24]. A baseline study showed that swimming crabs (Portunus trituberculatus) from the Shanghai market had varying degrees of Cd contamination [25]. Thus, there is great importance in assessing the Cd exposure in aquatic products.

Zhejiang Province is located on the southeast coast of China, with latitude from 27°02′ N to 31°11′ N and longitude from 118°01′ E to 123°10′ E, and is also home to 50.95 million people; this population not only consumes but also provides seafood for the rest of China and even abroad [26]. The objective of the present study was to evaluate the Cd health risk in aquatic foods based on food consumption data from residents in Zhejiang Province, China. The paper can offer several insights about Cd accumulation in aquatic products and serve as a basis to compare with other regions worldwide.

## 2. Materials and Methods

### 2.1. Sample Collection and Preparation

A total of 10,356 food samples were collected by professional samplers between 2018 and 2022 from shopping malls, seafood retail stores, and restaurants in 11 cities of Zhejiang Province, China. The 10,356 samples were separated into six food types: freshwater fish, marine fish, freshwater crustaceans, marine crustaceans, mollusks, and others. The samples were saved in a freezer at −18 °C and used experiments as quickly as possible.

### 2.2. Inductively Coupled Plasma Mass Spectrometry

The analysis of Cd in aquatic products was carried out based on the elemental analysis guide in the China Food Safety National Monitoring Manual and China National Food Safety Standard GB 5000.268-2016 [27]. A quantity of 0.2–0.5 g of each sample was placed in a microwave digestion inner tank, to which 5–10 mL of concentrated nitric acid was then added, and digestion was conducted according to the standard operation procedure of the microwave digestion instrument. In the meantime, blank and reference experiments were conducted. The Cd concentration was detected by inductively coupled plasma mass spectrometry (ICP-MS, 350D, Perkin Elmer, Inc., Shelton, CT, USA). The sample solutions, standards and reference materials were spiked with 50 ng/mL of the internal standard. The limit of detection (LOD) refers to the corresponding three times the value of the instrument background signal is generated by the matrix blank. The limits of quantification (LOQ) are determined by a signal-to-noise ratio of 10:1 or are approximated by multiplying the LOD by 3.3. The LOD for Cd by ICP-MS was 0.002 mg/kg, and the LOQ was 0.005 mg/kg. The recovery rate was ≥90% and the correlation coefficient R^2^ ≥ 0.995.

### 2.3. Food Consumption Data

An urban and rural population-based survey of food consumption data was administered in Zhejiang Province between 2015 and 2016. A total of 18,913 people from 22 townships were surveyed, covering all age groups (≥2 years). All participant’s information was collected, and they also were asked to complete a 24 h dietary recall during three consecutive days as accurately as possible. All the food consumed during the investigation were collected. We applied a stratified random sampling method over three consecutive days, including two working days and one weekend and excluding holidays. In the meantime, all participants’ body weight data were collected. For the consumption data, freshwater fish, marine fish, and marine crustaceans were the most consumed foods, with 16.487, 15.199, and 9.233 g/day, respectively (Table 1). The Ethics Review Committee of Zhejiang Provincial Center for Disease Control and Prevention (CDC) approved this survey, and verbal consent was obtained. Zhejiang CDC is authorized by the Zhejiang Provincial Government to collect and utilize food consumption information. All the participants were informed that they have the right to decline or terminate the study at any time during the interview. In addition, China’s National Center for Food Safety Risk Assessment (CFSA) was responsible for organizing and summarizing the food consumption survey, and our use of the data was verbally approved by CFSA.

### 2.4. Exposure Assessment

The consumption data of a particular food were combined with the Cd concentration and then divided by body weight to obtain the daily dietary exposure for each individual food item [28]. The average dietary exposure was represented by the mean, and 97.5th percentile (P97.5) was applied to represent the high exposure level. The average daily doses of Cd dietary exposure per kg of body weight were calculated by using the following formula:$$ADD=\frac{C\cdot IR}{BW}\cdot 30$$
where *ADD* is the average daily dose of Cd intake per kg body weight (mg/kg BW/month), *C* is the determined level of Cd in aquatic products, *IR* is the daily consumption of every aquatic food type, and *BW* is the body weight of the individual (kg). The estimated average monthly exposure level was compared with the PTMI of 0.025 mg/kg BW/month.

### 2.5. Health Risk Classification

A health risk classification method was used to accurately recognize the magnitude of health risks and provide scientific basis for the implementation of food safety risk management [29]. There were three stages. Firstly, the consequence severity was determined. Second, the possibility of adverse reactions when exposed to a poison was determined. Finally, the risk level was determined according to the risk matrix. Primarily, acute toxicity was expressed as “rat oral lethal Dose 50 (LD50)” (represented by “Ha”), and chronic toxicity was expressed by varied toxicity category points, including carcinogenicity, mutability, reproduction toxicity, developmental neurotoxicity, subchronic and chronic toxicity (represented by “Hb”), and the total point of consequence was the half the Ha and Hb. Next, the ratio average exposure to Cd in the target population (Pa) was used to ensure the “likelihood of adverse effects”, and the ratio of populations with individual exposure levels exceeding health guideline value was deemed as Pb to determine the likelihood caused by Cd, and the likelihood score was equal to half of the Pa and Pb scores combined. The final risk score combined the score of consequence with likelihood and shown in the risk matrix. The risk score was presented as three levels: low, medium, and high scores which were 1–4, 5–22, and 15–25, respectively. The different risk levels were colored green, yellow, and red.

### 2.6. Data Analysis

Based on the Reliable Evaluation of Low-Level Contaminations of Food report by Kulmbach in the GEMS/Food-EURO Second Workshop in 1995, he pointed out that the data sets (which had 60% or less data censored) using LOD_1/2_ for data reported as <LOD provided a reasonably effective estimate [30]. Therefore, the values of Cd concentration lower than LOD were deemed as LOD_1/2_ in the present study. All statistical analysis was conducted by IBM SPSS Statistics for Windows, version 25.0 (IBM Crop, Armonk, NY, USA). All of the results are expressed as the mean ± SD, and one-way analysis of variance (ANOVA) followed by Turkey’s test was used to evaluate the comparison between two groups. A value of *p* < 0.05 was considered statistically significant. Open source software QGIS (Quantum GIS version 3.22.9) was used to map the spatial distribution of Cd concentration in aquatic products among 11 cities from 2018 to 2022.

## 3. Results

### 3.1. Concentration of Cd in Aquatic Products

Cd concentrations were detected in 10,356 aquatic product samples from 2018 to 2022 and are summarized in Table 1. In total, 16.29% (1687 of 10,356) of the test samples were under the LOD. The mean Cd concentration in the six aquatic food groups was 0.699 mg/kg. Cd contents varied dramatically among food categories. The highest average concentration was 1.47 mg/kg for Cd in marine crustaceans. In addition, high Cd contents were found in mollusks (0.925 mg/kg) and others (0.180 mg/kg). Cd was present at low levels in freshwater fish, marine fish, and freshwater crustaceans, with means of 0.00988, 0.0417, and 0.0940 mg/kg, respectively.

In terms of temporal distribution, the highest Cd concentration was 0.869 mg/kg in 2020. Moreover, the average Cd concentrations showed a peak distribution from 2018 to 2022 (Figure 1a). The regional distribution of Cd concentration in aquatic products among 11 cities is shown in Figure 1b. The mean Cd concentrations in Zhoushan, Wenzhou, and Taizhou cities were the highest, with means of 1.157, 1.059, and 1.022 mg/kg, respectively. However, the cities of Shaoxing, Jinhua, and Huzhou had the lowest mean Cd concentrations of 0.230, 0.394, and 0.411 mg/kg, respectively.

### 3.2. Exposure Assessment

#### 3.2.1. Dietary Cd Exposure by Different Food Groups

The above formula was applied to evaluate the Cd exposure level. Mean exposure, P97.5, and the Cd contribution of different foods to dietary exposure are given in Table 2. The Cd average exposure of the general population in aquatic foods was 0.00951 mg/kg BW/month, which was equivalent to 38.04% of PTMI. Cd exposure in the high consumer (P97.5) of the general population was 0.0589 mg/kg BW/month, which was approximately 2.355 times the PTMI. The results indicated that marine crustaceans are the major Cd contributor to the Zhejiang population by combining the concentration and food consumption data in these six aquatic food types, corresponding to 82.7%. Other aquatic food category contributors ranged from 0.648% to 10.1%.

#### 3.2.2. Dietary Cd Exposure by Different Area

The regional Cd exposure distribution among 11 cities is shown in Table 3. In this study, the average Cd exposure levels of 11 cities in Zhejiang Province were 0.000254 to 0.0229 mg/kg BW/month. The cities of Zhoushan, Wenzhou, and Taizhou had the highest Cd exposure levels with 0.0229, 0.0182, and 0.0182 mg/kg BW/month, respectively. And all cities’ Cd exposure levels were below the PTMI.

#### 3.2.3. Cd Exposure by Age Subgroups

The aquatic product Cd exposures for all participants and different age–sex subgroups of the individuals are shown in Table 4. For adults, the average Cd exposure level was 0.00840 mg/kg BW/month between males and females, accounting for approximately 33.60% of the PTMI. Children aged 2–3 years were significantly more exposed to Cd than adults and young people but the Cd exposure levels of all age groups was under the PTMI (Table 4). For the high consumer group (P97.5), the Cd exposure levels of all age–sex groups were greater than the PTMI, ranging from 0.0493 (male adult) to 0.152 (children 2–3 years) mg/kg BW/month.

### 3.3. Health Risk Classification Assessment

We carried out health risk classification assessment in accordance with the above method. According to the score definitions and classification, the acute toxicity of LD50 was 2330 mg/kg, so the Ha score was set to two. Cd has chronic toxicity (category 1), so the Hb score was 4. Therefore, the overall score of consequence is three, indicating that the consequence level was “moderate”. According to our results, the ratio of average exposure to Cd in target population (Pa) accounted for 38.04% of the PTMI. Accordingly, the overall score of Pa was set to two. The proportion of individuals whose exposure level exceeded the PTMI was 9.42%. Therefore, the score of Pb was two. Therefore, the overall score of likelihood of adverse effect was two, and the likelihood level was “unlikely” in the risk matrix. Combining the levels of consequence and likelihood in the risk matrix, the horizontal axis and the vertical axis intersect at a yellow block (Figure 2). The final risk score is six, and the health risk level of Cd exposure in aquatic products in the Zhejiang population is medium.

## 4. Discussion

Dietary Cd intake has been a vital public health concern worldwide, and many studies have focused on the intake and source of dietary Cd [31,32]. This is the first assessment of Cd exposure in aquatic products in Zhejiang Province. The mean Cd concentration in the six aquatic food groups was 0.699 mg/kg, and the mean Cd exposure of the target population was 0.00951 mg/kg BW/month, which was below the PTMI of 0.025 mg/kg BW/month. According to the dietary Cd exposure assessment among the Chinese population in 2017, the result was lower than 0.0153 mg/kg BW/month [17]. In the Sixth China Total Diet Studies (TDSs) during 2016–2019, the mean Cd intake for Chinese adult males was 0.00826 mg/kg BW/month (0.0026–0.03002 mg/kg BW/month), which indicated that the mean Cd exposure in aquatic products in Zhejiang Province is at the national average level [33]. According to some previous reports, the mean concentration of Cd was 0.0085 mg/kg in Jilin Province and 0.159 mg/kg in Guangdong Province [34,35]. The average dietary exposures to Cd were 0.0184 mg/kg BW/month in Jiangxi Province and 0.0099 mg/kg BW/month in Shenzhen adults [36,37].

Cd dietary exposure shows that the Cd contribution of meat and vegetables has gradually increased over time, and aquatic foods have become one of the main sources of Cd exposure [38]. A heavy metal health risk assessment of aquatic products in the Chinese population revealed that the highest mean level of Cd was 0.431 mg/kg in shellfish and the lowest level was 0.006 mg/kg in fish [39]. A previous study showed that the level of Cd in aquatic products was less than 0.1 mg/kg and the highest average concentration was found at 0.196 mg/kg in swimming crabs in Zhejiang Province [40]. The health risk assessment from Taihu lake in China showed that the level of measured Cd in Taihu fish was moderate–low and that the consumption of aquatic foods from the lake was generally safe [41]. The results from Northeast China also indicated that the level of Cd in freshwater products was generally in an acceptable range [42]. However, seafood can be a prominent source of Cd. The bays in eastern China have been seriously polluted with heavy metals for a long time due to intensive anthropic pressure. In these pollutions and sediments, Cd poses a considerable ecological risk due to its high enrichment [43]. In our study, marine crustacean was the main Cd source of aquatic foods and contributed 81.66%. In general, the Cd concentration increases with the age/size of most species and tissues [44]. Among aquatic organisms, crustaceans stand out in the assessment of heavy metals, especially crabs, compared with fish and mollusks [45]. The main reason for crustaceans’ effective accumulation of heavy metals is the close contact of these benthic animals with polluted sediments and issues, such as bioturbation [46,47]. Most heavy metals accumulate in the crustacean liver, pancreas, and gills because the liver and pancreas are the main bioaccumulating organs and gills capture and accumulate metals through water [48].

Regional differences in Cd exposure level were observed in our study. Zhoushan, Wenzhou, and Taizhou cities reported the highest Cd concentrations with means of 1.157, 1.059, and 1.022 mg/kg, respectively. The mean Cd exposure levels of 11 cities in Zhejiang Province ranged from 0.000254 to 0.0229 mg/kg BW/month, and all cities’ Cd exposure levels did not exceed the PTMI. The Cd exposure levels of coastal cities were higher than those of other cities. Because deep-seabed mining contains various minerals and metals, with the decrease in land resources, deep-sea mining has drawn more increasing attention [49]. In coastal and estuary regions, Cd can enter the ocean from anthropogenic activities such as refining, copper and nickel smelting, fossil fuel combustion, and municipal wastes [50]. In addition, mounting evidence has demonstrated that ocean acidification is associated with metal pollution on coasts and estuaries. Ocean acidification aggravates the Cd toxicity to an environmentally relevant concentration, and the increased accumulation and altered subcellular distribution of Cd could lead to an increase in toxicity in marine products [51]. Moreover, climate change-related effects on the structure and function of ocean product webs, with consequent changes in contaminant transport, fate, and effects, are likely to have significant repercussions to marine products [52].

Furthermore, Cd exposure in aquatic food tended to decrease with age. Children (2–3 years old) were exposed to more Cd than young people (12–17 years old) and adults (over 18 years old). There are some plausible explanations as to why Cd exposure levels tend to decrease with age. Primarily, body weight is a huge factor since food consumption (kg/BW) was greater in children than in other subgroup populations [17], as children have the lowest body weight. Our results are consistent with the exposure assessment in the adult population and preschool children in the Republic of Serbia [53]. A risk assessment of dietary Cd exposure to Cd in residents of Guangzhou demonstrated that young people had lower margin of exposure (MOE) values (0.92) and indicated a higher risk, while those for the other age groups were all above 1.0 [18]. Additionally, young children and infants may be exposed to more Cd because they eat more food than is needed to maintain their weight (to realize growth and development), and therefore, Cd is more absorbable for them than adults [54]. Due to incomplete development of infants and young children, the developing brain is particularly vulnerable to permanent damage by exposure to environmental substances during the fragile window time. Therefore, high attention should be given to child development, and child health can be better protected by more preventative management of dietary and environmental chemicals.

There are some strengths in our study. Primarily, we applied the simple distribution method, which assessed Cd exposure based on food consumption and concentration. In terms of the results of health risk classification assessment, we concluded that the final risk score was 6, and the health risk level of Cd was medium. Application of the method in an actual case provided useful insights and could quickly determine the risk level of importance. The risk classification model was proposed by the China National Center for Food Safety Risk Assessment (CFSA) [29]. In addition, this is the first systematic evaluation of Cd exposure through aquatic products in Zhejiang Province. The overall risk of Cd exposure is manageable, and we paid more attention to the types of aquatic food and age subgroups. However, there are still several shortcomings. First, the food consumption data in 2015–2016 were old, and the dietary pattern in China has since changed due to the rapid development of the economy. Second, 3-day dietary records did not represent the true dietary Cd exposure during a prolonged period. In addition, the level of Cd entering the body was lower than that which the data detected due to the impact of cleaning, processing, cooking methods, and removing of inedible parts not being taken into consideration [21,55].

## 5. Conclusions

In summary, we systematically assess that the health risk from Cd exposure of Zhejiang provincial residents in aquatic products was low for the general population. We conclude that the risk of Cd intake through aquatic products is low among the general population, but that we should pay attention to marine crustaceans and young groups. Since the heavy metal contamination of aquatic foods is a severe public health issue, we recommend that it is necessary to further focus on the causes including industrial pollution, and geological and environmental protection. Future efforts can be undertaken to realize comprehensive assessments of the combined exposure and interaction of multiple pollutants.

## Figures and Tables

**Figure 1 foods-12-03094-f001:**
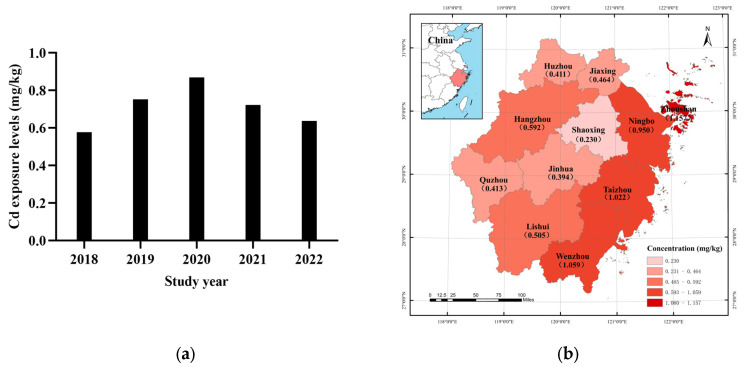
Cd concentration levels in aquatic products in 11 cities in Zhejiang Province, China, from 2018 to 2022. (**a**) Temporal distribution from 2018 to 2022; (**b**) temporal distribution in 11 cities.

**Figure 2 foods-12-03094-f002:**
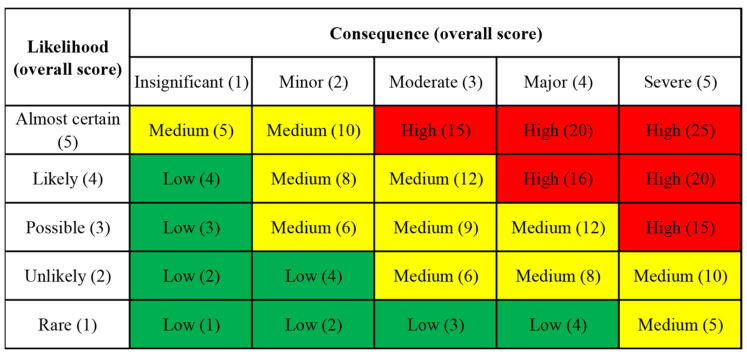
The consequence of health risk classification assessment.

**Table 1 foods-12-03094-t001:** Cd concentrations in the six aquatic food groups in Zhejiang Province, China.

Food Type	N	Cd Concentration (mg/kg)				Food Consumption (g/day)
		<LOD	Mean ± SD	Median	P97.5	
Freshwater fish	1113	677	0.00988 ± 0.0325	0.00150	0.0937	16.5 ± 26.3
Marine fish	832	227	0.0417 ± 0.236	0.00636	0.258	15.2 ± 29.7
Freshwater crustaceans	2255	544	0.0940 ± 0.394	0.0130	0.550	3.91 ± 11.8
Marine crustaceans	3247	116	1.47 ± 2.41	0.604	8.290	9.23 ± 18.8
Mollush	2249	60	0.925 ± 1.56	0.390	5.330	1.83 ± 6.76
Others	660	63	0.180 ± 0.957	0.0230	1.200	0.588 ± 4.59
Total	10,356	1687	0.699 ± 1.681	0.0582	5.200	47.3 ± 51.6

**Table 2 foods-12-03094-t002:** Cd exposure levels in aquatic foods and their contributions to dietary exposure level (%) in Zhejiang Province, China.

Food Type	Cd Exposure (mg/kg BW/month)			
	Mean ± SD	Median	P97.5	Contribution (%)
Freshwater fish	0.0000891 ± 0.000143	0	0.000470	0.937
Marine fish	0.000360 ± 0.000672	0	0.00210	3.79
Freshwater crustaceans	0.000214 ± 0.000735	0	0.00200	2.25
Marine crustaceans	0.00786 ± 0.0176	0	0.0538	82.7
Mollush	0.000960 ± 0.00373	0	0.0113	10.1
Others	0.0000616 ± 0.000495	0	0.000766	0.648
Total	0.00951 ± 0.0189	0.000901	0.0589	100

**Table 3 foods-12-03094-t003:** Cd exposure levels in aquatic products in 11 cities in Zhejiang Province, China.

Area	N	Exposure (mg/kg BW/month)		
		Mean ± SD	Median	P97.5
Hangzhou	1715	0.00216 ± 0.00632	0.000171	0.0201
Ningbo	2718	0.0148 ± 0.0193	0.00800	0.0662
Wenzhou	2672	0.0182 ± 0.0220	0.01220	0.0753
Jiaxing	1890	0.00618 ± 0.0152	0.000562	0.0457
Huzhou	932	0.00441 ± 0.0105	0.000497	0.0295
Shaoxing	889	0.00567 ± 0.0111	0.000559	0.0352
Jinhua	2759	0.00213 ± 0.00828	0.0000534	0.0236
Quzhou	863	0.000254 ± 0.00156	0.00000	0.000933
Zhoushan	889	0.0229 ± 0.0250	0.0170	0.0852
Taizhou	1825	0.0182 ± 0.0275	0.0103	0.0827
Lishui	1761	0.0039 ± 0.0185	0.000130	0.0345
Total	18,913	0.00951 ± 0.0189	0.000900	0.0589

**Table 4 foods-12-03094-t004:** Cd exposure levels in aquatic foods by age–sex groups of the population in Zhejiang Province, China.

Population Groups	N	Cd Exposure (mg/kg BW/month)		
		Mean ± SD	Median	P97.5
2–3 yr	307	0.0239 ± 0.0504 ^c,d,e,f^	0.00312	0.152
4–11 yr	1816	0.0161 ± 0.0315 ^c,e,f^	0.00141	0.0958
12–17 yr,male	504	0.0105 ± 0.0205 ^a,b^	0.000698	0.0730
12–17 yr,female	525	0.0107 ± 0.0198 ^a^	0.000757	0.0660
≥18 yr,male	7868	0.00816 ± 0.0143 ^a,b^	0.000824	0.0493
≥18 yr,female	7893	0.00864 ± 0.0158 ^a,b^	0.000901	0.0546
General population	18,913	0.00951 ± 0.0189	0.000901	0.0589

^a^ statistically significant difference between 2–3 yr and any other age–sex group at *p* < 0.01. ^b^ statistically significant difference between 4–11 yr and any other age–sex group at *p* < 0.01. ^c^ statistically significant difference between 12–17 yr(male) and any other age–sex group at *p* < 0.01. ^d^ statistically significant difference between 12–17 yr(female) and any other age–sex group at *p* < 0.01. ^e^ statistically significant difference between ≥18 yr(male) and any other age–sex group at *p* < 0.01. ^f^ statistically significant difference between ≥18 yr(female) and any other age–sex group at *p* < 0.01.

## Data Availability

The data used to support the findings of this study can be made available by the corresponding author upon request.

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
