# Peer review of "Cadmium Exposure in Aquatic Products and Health Risk Classification Assessment in Residents of Zhejiang, China"

_foods, 2023, doi:10.3390/foods12163094_

Round 1

Reviewer 1 Report

The manuscript is well written, and the expressive set-up is up to par. My comments to the authors are given below:

  1. Change the highlights so that you don't repeat the words from the title.
  2. The introduction part can be written better. I recommend including more recent works in this section.
  3. Rewrite lines 40–46.
  4. Materials and methods: please rewrite the Data analysis part.
  5. results: what is that "3.3" after table 3?
  6. The resolution of Figure 2 is very low. Please correct it.
  7. Discussion: This part is well written. But please recheck the formatting.
  8. Conclusions: Rewrite this section to better understand the main finding of your study.

Author Response

Dear reviewer:

Thank you for your letter and the reviewers’ comments on our manuscript entitled “Cadmium exposure in aquatic products and health risk classification assessment in residents of Zhejiang, China” (ID: foods-2559966). Those comments are very helpful for revising and improving our paper, as well as the important guiding significance to other research. We have studied the comments carefully and made corrections which we hope meet with approval. The revised manuscript with track changes please see the attachment. The main corrections are in the manuscript and the responses to the reviewers’ comments are as follows (the replies are highlighted in red ).

The manuscript is well written, and the expressive set-up is up to par. My comments to the authors are given below:

  1. Change the highlights so that you don't repeat the words from the title.Q1

Answer to Q1:

Thank you for your comments regarding our manuscript. We have corrected our Abstract part because we did not provide the highlight in our manuscript. We apologize for the repetition of the words in the title, abstract, and main text. Based on your comment, we have carefully checked the manuscript, and avoided the repetition.

  1. The introduction part can be written better. I recommend including more recent works in this section.Q2

Answer to Q2:

We feel great thanks for your professional review work of our manuscript. We apologize for the poor writings. As per your concern, we have added more recent works to our Introduction part. We have modified the statement as “Cd is extremely toxic heavy metal, and there is no specific, safe, and efficacious therapeutic management of Cd toxicity [3].” (Page 1, Line 35-36); “Cd toxicity as an important health hazard, which may compromise host defense against pathogenic microorganisms and homeostatic reparative activities. Cd exposure triggers oxidative stress, necroptosis, Th1/Th2 imbalance and promotes inflammation through the TNF-α/NF-κB pathway [9].” (Page 2, Line 40-44); “Due to the relatively high volatility, large ionic radius, and chemical speciation, Cd is susceptible to mobilization by anthropogenic and natural processes [11]. Cd in the environment can be enriched by foods and then transferred to human beings by the food chain. In 2011, the joint FAO/WHO Expert Committee on Food Additives (JECFA) formulated the health guideline value (HBGV) of Cd, which was that the daily intake of Cd for adults should not exceed 50 μg, and the provisional tolerable monthly intake (PTMI) for Cd was 0.025 mg/kg body weight (BW) [12]. The European Food Safety Commission (EFSA) set an HBGV of no more than 21.6 μg/day [13].” (Page 2, Line 46-53); “Generally, the quality of water is compromised due to the contamination of toxic metals, which impacts adversely on the health of humans, animals, and plants. A risk assessment of 87 Chinese lakes from 2009 to 2019 indicated that 5.6% of lake water and 88.2% of lake sediments were polluted by Cd [19]. The survey of 108 sediment samples from the Bohai Sea showed that the Cd concentrations were considerably higher than the average upper crust value and presented high potential ecological risk [20].” (Page 2, Line 65-70); We have also added additional supporting references.

  1. Bhattacharya S. The Role of Probiotics in the Amelioration of Cadmium Toxicity. Biological trace element research. 2020;197(2):440-4.
  2. Chen X, Bi M, Yang J, Cai J, Zhang H, Zhu Y, et al. Cadmium exposure triggers oxidative stress, necroptosis, Th1/Th2 imbalance and promotes inflammation through the TNF-α/NF-κB pathway in swine small intestine. Journal of hazardous materials. 2022;421:126704.
  3. Cullen JT, Maldonado MT. Biogeochemistry of cadmium and its release to the environment. Metal ions in life sciences. 2013;11:31-62.
  4. European Food Safety Authority. Cadmium in food: Scientific opinion of the panel on contaminants in the food chain. EFSA Journal, 2009, 980: 1-139, doi:10.2903/j.efsa.2009.980.
  5. Qin Y, Tao Y. Pollution status of heavy metals and metalloids in Chinese lakes: Distribution, bioaccumulation and risk assessment. Ecotoxicology and environmental safety. 2022;248:114293.
  6. Li H, Gao X, Gu Y, Wang R, Xie P, Liang M, et al. Comprehensive large-scale investigation and assessment of trace metal in the coastal sediments of Bohai Sea. Marine pollution bulletin. 2018;129(1):126-34.
  7. Rewrite lines 40–46.Q3

Answer to Q3:

Thank you very much for your helpful comments on our manuscript. According to your nice suggestion, we have modified the sentences in the revised manuscript as “ Due to the relatively high volatility, large ionic radius, and chemical speciation, Cd is susceptible to mobilization by anthropogenic and natural processes [11]. Cd in the environment can be enriched by foods and then transferred to human beings by the food chain. In 2011, the joint FAO/WHO Expert Committee on Food Additives (JECFA) formulated the health guideline value (HBGV) of Cd, which was that the daily intake of Cd for adults should not exceed 50 μg, and the provisional tolerable monthly intake (PTMI) for Cd was 0.025 mg/kg body weight (BW) [12]. The European Food Safety Commission (EFSA) set an HBGV of no more than 21.6 μg/day [13].” (Page 2, Line 46-53) Thank you again for your positive comment.

  1. Materials and methods: please rewrite the Data analysis part.Q4

Answer to Q4:

Thank you for your comments on our manuscript. According to your suggestion, we have modified the Data analysis part in the revised manuscript as “Based on the report Reliable Evaluation of Low-Level Contaminations of Food by Kulmbach in GEMS/Food-EURO Second Workshop in 1995, he pointed out that the data sets (which had 60% or less data censored) using LOD1/2 for data reported as 1/2 in the present study. All statistical analysis was conducted by IBM SPSS Statistics for Windows, version 25.0 (IBM Crop, Armonk, NY, USA). All of the results are expressed as the mean ± SD, and one-way analysis of variance (ANOVA) followed by Turkey’s test was used to evaluate the comparison between two groups. A value of P < 0.05 was considered statistically significant. Open source software QGIS (Quantum GIS version 3.22.9) was used to map the spatial distribution of Cd concentration in aquatic products among 11 cities from 2018 to 2022.” (Page 4, Line 163-173) And they were highlighted in red and underlined.

  1. results: what is that "3.3" after table 3? Q5

Answer to Q5:

We apologize for the mess and thank the reviewer for the reminder. We apologize for our carelessness. Based on your comments, we have checked the format of figures and tables in our revised manuscript. Thank you again for your comment.

  1. The resolution of Figure 2 is very low. Please correct it. Q6

Answer to Q6:

We thank the reviewer for pointing out this issue. We apologize for the low resolution of Figure 2 in our manuscript. As per your concern, we have submitted a more clearer Figure 2 in our revised manuscript, and we have ensured the resolution was 300 dpi. Thank you again for your comment.

  1. Discussion: This part is well written. But please recheck the formatting. Q7

Answer to Q7:

Thank you very much for your affirmation and approval to our work. Based on your comment, we have checked the revised manuscript carefully and made corresponding revisions including the formatting, typos, and grammatical errors. Thank you very much for your help with our work.

  1. Conclusions: Rewrite this section to better understand the main finding of your study. Q8

Answer to Q8:

Special thanks for your constructive comment. To address your concern, we have changed the Conclusion part: “In summary, we systematically assess the health risk from Cd exposure of Zhejiang provincial residents in aquatic products was low for the general population. We concluded that the risk of Cd intake through aquatic products is low among the general population, but we should pay attention to the marine crustaceans and young groups. Since heavy metal contamination of aquatic foods is a severe public health issue, we recommend that it is necessary to further focus on the causes including industrial pollution, geological and environmental protection. Future efforts can be taken on the combined exposure and interaction of multiple pollutants to realize comprehensive assessments.” (Page 9, Line 363-371)

Once again, thank you very much for your constructive comments and suggestions which would help us both in English and in depth to improve the quality of the paper.

Yours sincerely,

Yue He

Reviewer 2 Report

This study investigated the status of cadmium exposure due to food over a 3-day period among residents of Zhejiang Province. The sample size is sufficiently large and the results obtained are described as statistically useful.

The text is very readable and details the importance of risk assessment of exposure to cadmium. The discussion is also well developed.

I would like to request two points to be added to the text.

1) In all tables, state how the significant digits of the numerical values were determined.

2) For the opening sentence of 4. Discussion (L.225), provide several references.

Author Response

Dear reviewer:

Thank you for your letter and the reviewers’ comments on our manuscript entitled “Cadmium exposure in aquatic products and health risk classification assessment in residents of Zhejiang, China” (ID: foods-2559966). Those comments are very helpful for revising and improving our paper, as well as the important guiding significance to other research. We have studied the comments carefully and made corrections which we hope meet with approval. The revised manuscript with track changes please see the attachment. The main corrections are in the manuscript and the responses to the reviewers’ comments are as follows (the replies are highlighted in red). The responses to your comments are as follows:

This study investigated the status of cadmium exposure due to food over a 3-day period among residents of Zhejiang Province. The sample size is sufficiently large and the results obtained are described as statistically useful.

The text is very readable and details the importance of risk assessment of exposure to cadmium. The discussion is also well developed.

I would like to request two points to be added to the text.

  1. In all tables, state how the significant digits of the numerical values were determined.Q1

Answer to Q1:

Thank you very much for your careful review and reminder. According to your suggestion, we have corrected the significant digits of the numerical values in all tables, and three significant digits were preserved. We appreciate your valuable comments and suggestions for improving our manuscript.

2.For the opening sentence of 4. Discussion (L.225), provide several references. Q1

Answer to Q2:

Special thanks for your constructive comment. We apologize for the limited literature review. According to your suggestion, we have modified the opening sentence of Discussion as “Dietary Cd intakes have been a vital public health concern worldwide, and many studies have focused on the intake and source of dietary Cd [30-31].” (Page 8, Line 274-275) and added several references.

  1. Satarug S, Vesey DA, Gobe GC. Current health risk assessment practice for dietary cadmium: Data from different countries. Food and chemical toxicology : an international journal published for the British Industrial Biological Research Association. 2017;106(Pt A):430-45.
  2. Ferrari P, Arcella D, Heraud F, Cappé S, Fabiansson S. Impact of refining the assessment of dietary exposure to cadmium in the European adult population. Food additives & contaminants Part A, Chemistry, analysis, control, exposure & risk assessment. 2013;30(4):687-97.

Once again, thank you very much for your constructive comments and suggestions which would help us both in English and in depth to improve the quality of the paper.

Yours sincerely,

Yue He

Reviewer 3 Report

Comments are given in the attached file

No comment

Author Response

Dear reviewer:

Thank you for your letter and the reviewers’ comments on our manuscript entitled “Cadmium exposure in aquatic products and health risk classification assessment in residents of Zhejiang, China” (ID: foods-2559966). Those comments are very helpful for revising and improving our paper, as well as the important guiding significance to other research. We have studied the comments carefully and made corrections which we hope meet with approval. The revised manuscript with track changes please see the attachment. The main corrections are in the manuscript and the responses to the reviewers’ comments are as follows (the replies are highlighted in red ).

  1. 6 should be replaced with six. (Line 14)Q1

Answer to Q1:

We agree with this suggestion and have modified the Abstract and replaced 6 with six. Thank you for your comments for our manuscript.

  1. PTMI: For the first appearance, abbreviated terms should be competently defined. (Line 20)Q2

Answer to Q2:

We appreciate your valuable comments and suggestions for improving our manuscript. We have added the complete definition of PTMI when it first appeared in the Abstract. Thank you very much for your help.

  1. needs space between the last character and brackets (Line 32)Q3

Answer to Q3:

Thank you very much for your helpful comments on our manuscript. According to your point, we have added all spaces between the last character and brackets in our revised manuscript. Thank you again for your positive comment.

  1. You could mention the national acceptable limits of Chinese authorities for Cd along the other world standards. (Line 51-52)Q4

Answer to Q4:

Special thanks for your constructive comment. According to your suggestion, we have tried our best to collect the acceptable limit for different authorities for Cd. We found that the limited value of Chinese national standards was from 0.003 mg/kg in natural mineral water to 3 mg/kg in sea crab. And we mentioned some other world standards, including European Union, Australia, and the international codex alimentarius commission (CAC). We have rewritten this sentence as “In China, especially the southern area, some foods contain Cd that exceeds the Chinese food limits, which is limited from 0.003 mg/kg in natural mineral water to 3 mg/kg in sea crab [16,17]. Compared to other standard worldwide, the food limits of Cd are 0.005-3 mg/kg in European Union, 0.05-2.5 mg/kg in Australia, and 0.003-2 mg/kg in international codex alimentarius commission (CAC), respectively.” (Page 2, Line 58-62) in our revised manuscript. We appreciate your help and would like to make other modifications that you suggest. Thank you very much for your help.

  1. Have the authors checked the lliterature for any possible effect of processing the seafoods (such as cooking or boiling) on the reduction of Cd content? How you can explain the confounding effect of ignoring this? (Line 74-80)Q5

Answer to Q5:

Thank you for your comment and professional review. For the effect of processing the seafoods (such as cooking or boiling), we pointed out “In addition, the level of entering the body was lower than the data we detected due to the impact of cleaning, processing, cooking methods and removing inedible parts not being taken into consideration [21, 54].” (Page 9, Line 352-355) in our Discussion part. As you concern, we thought the confounding effect on Cd content was not neglected. We apologize for the limited literature review, and we have added the literature to the revised manuscript:

  1. Schmidt, L.; Novo, D.R.; Druzian, G.T.; Landero, J.A.; Caruso, J.; Mesko, M.F., et al. Influence of culinary treatment on the concentration and on the bioavailability of cadmium, chromium, copper, and lead in seafood. Journal of trace elements in medicine and biology : organ of the Society for Minerals and Trace Elements (GMS) 2021, 65, 126717, doi:10.1016/j.jtemb.2021.126717.
  2. 54. Perugini M, Visciano P, Manera M, Abete MC, Tarasco R, Amorena M. Lead, cadmium and chromium in raw and boiled portions of Norway lobster. Food additives & contaminants Part B, Surveillance. 2014;7(4):267-72.
  1. Is there any data regarding the sensitivity, LOQ, LOD, recovery (%) and R2(R square)? need to include in the Materials and Methods section (Line 90)Q6

Answer to Q6:

We feel great thanks for your professional review work of our manuscript. As per your concerned, we have made some corrections to our previous draft. Thank you for your comment. We have added the description to the revised manuscript rewritten this sentence in Material and Methods section: “The limit of dection (LOD), refers to the corresponding 3 times the value of the instrument background signal generated by the matrix blank. The limits of quantification (LOQ) is determined by a signal-to-noise ratio of 10:1, or approximated by multiplying the LOD by 3.3. The LOD for Cd by ICP-MS was 0.002 mg/kg, and the LOQ was 0.005 mg/kg. The recovery rate was ≥ 90% and the correlation coefficient R2 ≥ 0.995.” (Page 3, Line 108-112)

  1. Provide data in this section. (Line 91)Q7

Answer to Q7:

Thanks for your nice suggestion. We apologize for this carelessness. Accordingly, we have added the data in the revised manuscript as “For the consumption data, freshwater fish, marine fish, and marine crustaceans were the most consumed foods, with 16.487, 15.199, and 9.233 g/day, respectively (Table 1).” (Page 3, Line 121-123) to address the reviewer’s point.

  1. As you are aware, nowadays the mixed and simultaneous contaminants are receiving attention due their importance in the complex matrices and their potential mutual interaction. Why the authors did not try to design such a study?(Line 310) Q8

Answer to Q8:

Thank you again for your positive comments on our manuscript. As you concern, we also paid an attention to the complex matrices and potential mutual interaction in the mixed and simultaneous contaminants. Your suggestions provide a very good idea and direction for our next research. The study of the combined exposure and interaction of multiple pollutants in the complex mechanism requires more systematic and complex hypotheses and models, and we are in the process of further learning and exploration. According to your points, we have modified the revised manuscript as “Future efforts can be taken on the combined exposure and interaction of multiple pollutants to realize comprehensive assessments.” (Page 9, line 369-371) in Conclusion part. Thanks again for your suggestion.

  1. "Bulletin of Environmental Contamination and Toxicology"Format all the references like this. (Line 335)Q9

Answer to Q9:

Special thanks for your constructive comment. To address your concern, we have checked the format of all references and changed the Reference part. We appreciate your valuable comments and suggestions for improving our manuscript.

Once again, thank you very much for your constructive comments and suggestions which would help us both in English and in depth to improve the quality of the paper.

Yours sincerely,

Yue He

Reviewer 4 Report

The manuscript titled “Cadmium exposure in aquatic products and health risk classification assessment in residents of Zhejiang, China” By Hexiang Zhang et al., submitted to “Foods” journal aims to evaluate the health risk of the Cd in aquatic foods based on food consumption data from residents of Zhejiang Province, China.

Some comments are raised. Please go through consider them in your revision.

1.      corresponding to 81.66% line #19,  in what sense? explain

2.      PTMI, write in full form at first appearance. Line #20

3.      Cadmium (Cd) don’t need to abbreviate again. Already did in abstract?

4.      Line 30 add space between citation and text.

5.      Line 34 again same typo mistake add space between citation and text.

6.      Add reference for  LOD were assumed to be LOD/2 and justify?

7.      Line 235 typo mistake

8.      Add space between citation and text.

9.      Why CD was low in fresh water?

10.  Add limitation and strengths of the research in appropriate manner.

11.  There are several spelling, punctuation and English grammar errors throughout the manuscript.

Author Response

Dear reviewer:

Thank you for your letter and the reviewers’ comments on our manuscript entitled “Cadmium exposure in aquatic products and health risk classification assessment in residents of Zhejiang, China” (ID: foods-2559966). Those comments are very helpful for revising and improving our paper, as well as the important guiding significance to other research. We have studied the comments carefully and made corrections which we hope meet with approval. The revised manuscript with track changes please see the attachment. The main corrections are in the manuscript and the responses to the reviewers’ comments are as follows (the replies are highlighted in red ).

The manuscript titled “Cadmium exposure in aquatic products and health risk classification assessment in residents of Zhejiang, China” By Hexiang Zhang et al., submitted to “Foods” journal aims to evaluate the health risk of the Cd in aquatic foods based on food consumption data from residents of Zhejiang Province, China.

Some comments are raised. Please go through consider them in your revision.

  1. corresponding to 81.66% line #19,  in what sense? Explain Q1

Answer to Q1:

Thank you for your comments and careful review. We thank the reviewer for pointing out this issue. In our study, we separately analyzed each aquatic food group for the general population, and the contribution to Cd exposure was shown in Table 2. By combining the data of Cd content and food consumption, we concluded marine crustaceans are the major Cd contributor to the Zhejiang population, corresponding to 81.66%. Other food group contributors ranged from 0.65% to 9.80%.

  1. PTMI, write in full form at first appearance. Line #20Q2

Answer to Q2:

Thank you for your comments and careful review. We have added the full form of PTMI in our revised manuscript. Thank you very much for your help with our work.

  1. Cadmium (Cd) don’t need to abbreviate again. Already did in abstract? Q3

Answer to Q3:

We feel great thanks for your professional review work of our manuscript. As per your concerned, we have obtained the full form of Cadmium in Introduction part according to submission requirements: Abbreviations should be defined the first time they appear in each of three sections: the abstract; the main text; the first figure or table. Thank you again for your comment.

  1. Line 30 add space between citation and text. Q4

Answer to Q4:

Thank you for your comment and professional review on our manuscript. According to your suggestion, we have carefully scrutinized the manuscript and added all spaces between the last character and brackets in our revised manuscript. Thank you again for your positive comment.

  1. Line 34 again same typo mistake add space between citation and text. Q5

Answer to Q5:

We apologize for the typo mistakes of our manuscript. Thank you very much for your helpful comment. In our revised manuscript, we have added all space between citation and text. Moreover, the added spaces are highlighted within the document in red and underlined. Thank you very much for your help.

  1. Add reference for  LOD were assumed to be LOD/2 and justify? Q6

Answer to Q6:

Thank you for your comments for our manuscript. Based on your suggestion, we have modified the sentence as “Based on the report Reliable Evaluation of Low-Level Contaminations of Food by Kulmbach in GEMS/Food-EURO Second Workshop in 1995, he pointed out that the data sets (which had 60% or less data censored) using LOD1/2 for data reported as ”(Page 4, Line 163-166) in our revised manuscript, and we also have added the reference:

  1. GEMS/Food-EUROS, editor Reliable Evaluation of Low-Level Contamination of Food. Second Workshop GEM/Food-EUROS 26-27 May Kulmbach, Federal Republic of Germany, 1995; 1995.
  2. Line 235 typo mistake Q7

Answer to Q7: Thank you for your critical comments and suggestions, with which we totally agree. We apologize for our carelessness. Based on your comments, we have checked the sentence and have corrected the mistake. Thank you again for your helpful comment.

  1. Add space between citation and text.Q8

Answer to Q8:

We apologize for the mess and thank the reviewer for the reminding. According to your nice suggestion, we have checked carefully the space and corrected all typo mistakes in our revised manuscript, and they were highlighted in red and underlined.

  1. Why CD was low in fresh water? Q9

Answer to Q9:

Thank you for your comment and professional review. In the Discussion part, we pointed out “The health risk assessment from Taihu lake in China showed that the level of measured Cd in Taihu fish was moderate-low and that the consumption of aquatic foods from the lake was generally safe [40]. The results from Northeast China also indicated that the level of Cd in freshwater products was generally in an acceptable range [41]. However, seafood can be a prominent source of Cd. The bays in eastern China have been seriously polluted with heavy metals for a long time due to intensive anthropic pressure. In these pollutions and sediments, Cd posed a considerable ecological risk due to its high enrichment [42].” (Page 8, Line 296-303) Our results are consistent with the above studies, and it is reasonable to assume that Cd is lower in freshwater products than in seafood. Thank you again for your comment.

  1. Add limitation and strengths of the research in appropriate manner. Q10

Answer to Q10:

Thank you again for your positive comments on our manuscript. According to your points, we have added the strengths and limitation into the Discussion section. The revised sentences are highlighted within the document in red text:

There are some strengths in our study. Primarily, we applied the simple distribution method, which assessed Cd exposure based on food consumption and concentration. In terms of the results of health risk classification assessment, we concluded that the final risk score was 6, and the health risk level of Cd was medium. Application of the method in an actual case provided useful insights and could quickly determine the risk level of importance. The risk classification model was proposed by the China National Center for Food Safety Risk Assessment (CFSA) [28]. In addition, this is the first systematic evaluation of Cd exposure of aquatic products in Zhejiang Province. The overall risk of Cd exposure is manageable, and we paid more attention to the types of aquatic food and age subgroups. However, there are still several shortcomings. First, the food consumption data in 2015-2016 were old, and the dietary pattern in China had changed due to the rapid development of the economy. Second, 3-day dietary records did not represent the true dietary Cd exposure during a prolonged period. In addition, the level of entering the body was lower than the data we detected due to the impact of cleaning, processing, cooking methods and removing inedible parts not being taken into consideration [21, 54].” (Page 9, Line 346-361)

  1. There are several spelling, punctuation and English grammar errors throughout the manuscript. Q11

Answer to Q11:

We thank the reviewer for pointing out this issue. We apologize for the spelling, punctuation and English grammar errors in our manuscript. We have carefully checked the manuscript, and made corresponding revisions including some typos, grammatical errors, and long sentences. Besides, we have made our manuscript polished by a professional editing company, and the article has been revised by a native English speaker. We are happy to edit the text further, based on helpful comments from the reviewers.

Once again, thank you very much for your constructive comments and suggestions which would help us both in English and in depth to improve the quality of the paper.

Yours sincerely,

Yue He

Round 2

Reviewer 3 Report

Need minor revision

Need minor revision, format references according to the Journal format.

Reviewer 4 Report

Accept in present form